# RETHINKING ONE-VS-THE-REST LOSS FOR INSTANCE-DEPENDENT COMPLEMENTARY LABEL LEARNING

## ABSTRACT

Complementary Label Learning (CLL) is a typical weakly supervised learning protocol, where each instance is associated with one complementary label, which specifies the class that the instance does not belong to. Existing CLL methods assume that the complementary label is sampled uniformly from all non-ground-truth labels, or from a biased probability depending on the ground-truth label. However, these assumptions are normally unrealistic, for example, an annotator tends to choose a label that is largely irrelevant to the instance to avoid mistaking the ground-truth label as the complementary one. Therefore, in this paper, we introduce instance-dependent CLL (IDCLL), where non-ground-truth labels that are less relevant to the instances are more likely to be selected as the complementary ones. Accordingly, we present our generation process for instance-dependent complementary label and observe that directly applying existing CLL methods to IDCLL results in poor performance. We further empirically analyze this phenomenon and identify: Existing methods exhibit a decline in their capacity to share complementary labels under the instance-dependent setting, resulting in small logit margins, thus difficult to identify ground-truth labels. To address this problem, we introduce *complementary logit margin loss* (CLML) and demonstrate CLML can enhance the capacity to share complementary labels. Additionally, we propose a novel form of the complementary one-vs-the-rest loss (COVR) as the surrogate loss for CLML, and provide theoretical proof that COVR can decrease CLML to a greater extent compared to existing CLL methods. The estimation error bound of the proposed COVR is also theoretically characterized. Extensive experiments conducted on benchmark datasets demonstrate the superiority of the proposed method compared to the existing CLL methods under our instance-dependent setting.

## 1 INTRODUCTION

In ordinary supervised classification, each instance is specified to the class it belongs to (Xue & Hauskrecht, 2019). However, accurately annotating large-scale datasets in a fully supervised manner can be both time-consuming and costly. To overcome this problem, weakly supervised learning protocols have gained increasing attention in recent years, including partial label learning (Tian et al., 2023; Wang et al., 2022), semi-supervised learning (Kou et al., 2023; Guo et al., 2022), noisy label learning (Natarajan et al., 2013; Li et al., 2022), and positive-unlabeled learning (Elkan & Noto, 2008; Wilton et al., 2022).

Particularly, in this paper, we consider another weakly supervised learning protocol called complementary label learning (CLL) (Ishida et al., 2017; 2019), where each instance is only associated with a complementary label to specify a class that the instance *does not* belong to. Then, CLL aims to learn a classifier to predict the ground-truth label for each instance by leveraging the complementary labels. Existing CLL methods primarily focus on two assumptions for the generation of the complementary label: the uniform assumption and the biased one. The uniform assumption assumes that each complementary label is sampled *uniformly* from all labels except the ground-truth label (Ishida et al., 2017; 2019; Chou et al., 2020), while the biased assumption considers that the complementary label is selected from a biased probabilities only depending on the ground-truth label (Yu et al., 2018). In other words, traditional CLL methods consider that the selection of complementary labels is *independent* of the instance.

Nevertheless, in real-world scenarios, the selection of complementary labels is mostly instance-dependent: An annotator tends to choose a complementary label that is largely unrelated to the instance, in order to ensure the correctness of the selection. Consequently, in this paper, we propose a realistic instance-dependent assumption:

*The non-ground-truth labels with lower relevance to the instance are more likely to be chosen as the complementary labels than the ones with higher relevance.*

Based on this assumption, we formally introduce the protocol of instance-dependent complementary label learning (IDCLL), which assumes that the selection of each instance-dependent complementary label (IDCL) explicitly/implicitly depends on the instance. In particular, to facilitate the generation of IDCL, we propose a systematic mechanism named "Min$k$" that utilizes a selection probability $\bar{p}(\bar{Y}|Y, X)$, which indicates the likelihood of selecting a non-ground-truth label as the complementary one based on its relevance to the instance. Specifically, a pre-trained model is used to obtain the selection probability and $k$ non-ground-truth labels with the highest selection probabilities are chosen. Subsequently, we randomly designate one label as the IDCL from the $k$ selected labels. Moreover, we present the definition of *complementary label distribution* and compare its difference between uniform and instance-dependent settings.

A straightforward attempt to solve IDCLL is to use existing CLL methods (Ishida et al., 2017; Yu et al., 2018; Chou et al., 2020) employed for instance-independent settings. However, our empirical study suggests that directly applying existing CLL methods to IDCLL leads to poor performance. In order to further investigate the underlying reason, we conduct in-depth investigation on existing CLL losses and explain why these methods work well under uniform setting. We first introduce the share complementary label hypothesis and logit margin loss (LML), and provide insight into the root cause of the poor performance: The sparser complementary label distribution under instance-dependent settings diminishes the capacity of existing CLL methods to effectively share complementary labels, resulting in small logit margin losses and making it challenging to disambiguate for the potential ground-truth labels. To tackle this challenge, we introduce complementary logit margin loss (CLML) and demonstrate that minimizing CLML can enhance model's capacity to share complementary labels. We also tailor a novel complementary one-versus-rest loss (COVR) as a surrogate loss to CLML. Our empirical and theoretical analysis has verified that the proposed COVR significantly enhances the capacity to share complementary labels and decreases CLML, which ultimately leads to an improved performance in identifying the ground-truth labels.

Our contributions are summarized in the following four levels:

- Problem setting level: We for the first time introduce the concept of IDCLL and provide a systematic mechanism for generating IDCLs according to our assumption.

- Empirical study level: We demonstrate the poor performance of existing CLL methods under the instance-dependent setting and conduct thorough empirical analysis to identify the underlying cause, i.e., the weaken capacity to share complementary labels.

- Methodology level: To enhance the capacity to share complementary label, we accordingly introduce CLML and propose COVR as the surrogate loss to CLML.

- Experimental level: We conduct extensive experiments on benchmark datasets to verify the effectiveness and superiority of the proposed method under different instance-dependent settings.

## 2 RELATED WORK

**Complementary label learning (CLL)** To solve CLL under uniform assumption, an unbiased risk estimator (URE) was derived using a specific complementary loss function (e.g., one-versus-all or pairwise comparison) that satisfies a symmetric condition (Ishida et al., 2017). Additionally, a more general URE for arbitrary loss functions and models was derived: To alleviate the overfitting issue of this URE in practice, non-negative correction and gradient ascent methods were further proposed (Ishida et al., 2019). However, URE suffers from huge gradient variance and results in unsatisfactory performance in practice. In order to mitigate this issue, the surrogate complementary loss framework (SCL) was proposed to reduce the gradient variance (Chou et al., 2020). Moreover, several promising approaches have been proposed to solve CLL, such as using weighted complementary loss (Gao & Zhang, 2021) and introducing partial-output regularization (Liu et al., 2023). Different from

the uniform assumption, biased CLL was considered where the selection of complementary labels depends on a biased probabilities (Yu et al., 2018). We should emphasize that IDCLL is different from biased CLL and a comprehensive understanding of the distinctions can be found in Appendix G.

**Instance-dependent weakly supervised learning** Instance-dependent assumption has been introduced into two weakly supervised learning protocols: Instance-dependent partial label learning (IDPLL) and instance-dependent label noise learning (IDN). IDPLL assumes that the labels with high relevance to the instance are more likely to be selected in the candidate set (Xu et al., 2021). Recovering the latent label distribution (Xu et al., 2021), performing Maximum A Posterior (MAP) based on an explicitly model generation process of candidate labels (Qiao et al., 2023), and purifying the candidate set with dynamic thresholds (Xu et al., 2022) has been proposed to solve IDPLL. Similarly, IDN assumes that poor quality or ambiguous instances in real-world datasets are more likely to be mislabeled (Garg et al., 2023), including the instance-dependent dichotomous label noise (Menon et al., 2018), bounded instance-dependent label noise (Cheng et al., 2020), and instance-dependent noise with confidence scores (Berthon et al., 2021). Moreover, several work directly estimated the transition matrix to solve IDN (Yang et al., 2022; Cheng et al., 2022; Yao et al., 2021; Xia et al., 2020). Our work is the first to introduce the instance-dependent assumption to CLL.

## 3 PRELIMINARIES

Different from supervised classification, each instance is only provided with *single* complementary label in CLL. Let $\bar{\mathcal{D}} = \{(\boldsymbol{x}_i, \bar{y}_i)\}_{i=1}^n$ denotes the complementary dataset, sampled from an unknown probability distribution $\bar{p}(\boldsymbol{x}, \bar{y})$, where $\bar{y}_i \in \mathcal{Y} \backslash \{y_i\}$ is the complementary label of the instance $\boldsymbol{x}_i$. The goal of CLL is to learn a DNN classifier $h_{\boldsymbol{\theta}}(\boldsymbol{x}_i) : \mathcal{X} \to \mathcal{Y}$, which is expected to predict the ground-truth label of an input $\boldsymbol{x}_i$:

$$h_{\boldsymbol{\theta}}(\boldsymbol{x}_i) = \underset{k \in \{1,2,...,K\}}{\operatorname{argmax}} f_k(\boldsymbol{x}_i, \boldsymbol{\theta}), \quad f_k(\boldsymbol{x}_i, \boldsymbol{\theta}) = e^{g_k(\boldsymbol{x}_i, \boldsymbol{\theta})} / \sum_{j=1}^{K} e^{g_j(\boldsymbol{x}_i, \boldsymbol{\theta})}, \quad (1)$$

where $\boldsymbol{g}(\boldsymbol{x}_i, \boldsymbol{\theta}) = [g_1(\boldsymbol{x}_i, \boldsymbol{\theta}), g_2(\boldsymbol{x}_i, \boldsymbol{\theta}), .., g_K(\boldsymbol{x}_i, \boldsymbol{\theta})]^T$ and $g_k(\boldsymbol{x}_i, \boldsymbol{\theta})$ is the $k$-th logit of the model with respect to the instance $\boldsymbol{x_i}$. Meanwhile, $\boldsymbol{f}(\boldsymbol{x}_i, \boldsymbol{\theta}) = [f_1(\boldsymbol{x}_i, \boldsymbol{\theta}), f_2(\boldsymbol{x}_i, \boldsymbol{\theta}), .., f_k(\boldsymbol{x}_i, \boldsymbol{\theta})]^T$ and $f_k(\boldsymbol{x}_i, \boldsymbol{\theta})$ is the predicted probability of $\boldsymbol{x}_i$ belonging to class $k$, i.e., $p_{\boldsymbol{\theta}}(Y = k | X = \boldsymbol{x}_i)$.

A commonly used CLL method based on DNNs is the Surrogate Complementary Loss (SCL) framework(Chou et al., 2020), which defines a novel Complementary 0-1 Loss:

$$\bar{\ell}_{01}(\bar{y}, h_{\boldsymbol{\theta}}(\boldsymbol{x})) = \mathbb{1}(h_{\boldsymbol{\theta}}(\boldsymbol{x}) = \bar{y}), \quad (2)$$

where $\mathbb{1}$ is the indicator function. Based on the Complementary 0-1 Loss, the expected complementary classification risk of classifier $h_{\boldsymbol{\theta}}$ on $\bar{p}(\boldsymbol{x}, \bar{y})$ can be defined as:

$$\bar{R}(h_{\boldsymbol{\theta}}; \bar{\ell}_{01}) = \mathbb{E}_{\bar{p}(\boldsymbol{x}, \bar{y})}[\mathbb{1}(h_{\boldsymbol{\theta}}(\boldsymbol{x}) = \bar{y})]. \quad (3)$$

Consequently, the expected complementary classification risk $\bar{R}(h_{\boldsymbol{\theta}}; \bar{\ell}_{01})$ is an Unbiased Risk Estimator (URE) of the expected risk of supervised classification $R(h_{\boldsymbol{\theta}}; \ell_{01})$ (Chou et al., 2020):

$$R(h_{\boldsymbol{\theta}}; \ell_{01}) = (K-1)\bar{R}(h_{\boldsymbol{\theta}}; \bar{\ell}_{01}). \quad (4)$$

In other words, minimizing the empirical complementary 0-1 risk is equivalent to the supervised learning. In order to mitigate the overfitting issue and improve traditional URE-based methods, Chou et al. (2020) proposed to use negative learning loss (SCL_NL) as the Surrogate Complementary Loss:

$$\bar{\ell}_{\text{SCL\_NL}}(\bar{y}, \boldsymbol{f}(\boldsymbol{x})) = -\log(1 - f_{\bar{y}}(\boldsymbol{x})) = -\log(1 - e^{g_{\bar{y}}(\boldsymbol{x})} / \sum_{j=1}^{K} e^{g_j(\boldsymbol{x})}). \quad (5)$$

## 4 INSTANCE-DEPENDENT COMPLEMENTARY LABEL LEARNING

### 4.1 GENERATION PROCESS FOR IDCLS

In this section, we introduce the generation process for IDCLs based on the instance-dependent assumption. In general, we select the IDCL for each instance $\boldsymbol{x}$ according to the probability $\bar{p}(\bar{Y} =$

Table 1: Classification accuracy (mean±std) of each baseline approach on benchmark datasets under uniform and "Min3" settings, respectively. The symbol ↓ indicates a decrease in classification accuracy for these approaches under the "Min3" setting compared to the uniform setting.

| Dataset | MNIST | | Kuzushiji-MNIST | | Fashion-MNIST | | CIFAR-10 | | SVHN | |
|---|---|---|---|---|---|---|---|---|---|---|
| Setting | Uniform | Min3 | Uniform | Min3 | Uniform | Min3 | Uniform | Min3 | Uniform | Min3 |
| W_Loss | $97.14 \pm 0.28$ | $46.19 \pm 4.56 \downarrow$ | $77.01 \pm 1.16$ | $46.73 \pm 3.94 \downarrow$ | $83.42 \pm 0.66$ | $42.61 \pm 6.14 \downarrow$ | $72.11 \pm 1.14$ | $45.57 \pm 0.63 \downarrow$ | $79.54 \pm 0.21$ | $38.46 \pm 3.13 \downarrow$ |
| Forward | $98.00 \pm 0.07$ | $51.82 \pm 6.81 \downarrow$ | $78.05 \pm 0.74$ | $53.69 \pm 4.51 \downarrow$ | $85.16 \pm 0.32$ | $39.54 \pm 0.13 \downarrow$ | $76.56 \pm 0.97$ | $45.12 \pm 2.03 \downarrow$ | $87.22 \pm 2.43$ | $31.43 \pm 2.86 \downarrow$ |
| SCL_NL | $98.06 \pm 0.11$ | $55.56 \pm 6.91 \downarrow$ | $78.21 \pm 0.35$ | $51.33 \pm 1.76 \downarrow$ | $85.18 \pm 0.31$ | $39.53 \pm 0.08 \downarrow$ | $74.94 \pm 4.06$ | $38.40 \pm 0.34 \downarrow$ | $82.32 \pm 4.72$ | $32.97 \pm 2.18 \downarrow$ |
| SCL_EXP | $97.73 \pm 0.15$ | $33.78 \pm 4.03 \downarrow$ | $76.86 \pm 0.36$ | $42.78 \pm 4.20 \downarrow$ | $84.70 \pm 0.13$ | $37.06 \pm 1.60 \downarrow$ | $74.02 \pm 0.26$ | $30.78 \pm 0.94 \downarrow$ | $79.58 \pm 5.06$ | $20.59 \pm 0.06 \downarrow$ |

$\bar{y}|Y=y, X=\boldsymbol{x})$, which reflects the likelihood of $\bar{y}$ being selected as the IDCL for $\boldsymbol{x}$. Recent work suggests that the prediction probability of a DNN classifier, i.e., $p(Y|X)$, indicates the label's relevance to the corresponding instance (Wu et al., 2018). Specifically, lower prediction probability means less association between the label and instance. Hence, we consider to utilize the prediction probability of a pre-trained model to generate the non-noisy IDCL by introducing the following assumption (Gao & Zhang, 2021):

**Assumption 1.** *For an instance $\boldsymbol{x}$ with ground-truth $y$, the selection probability $\bar{p}(\bar{Y}=j|Y=y, X=\boldsymbol{x}) = \frac{\exp(1-p(Y=j|X=\boldsymbol{x}))}{\sum_{k \neq y} \exp(1-p(Y=k|X=\boldsymbol{x}))}, j \in \mathcal{Y} \backslash \{y\}$, and $\bar{p}(\bar{Y}=y|Y=y, X=\boldsymbol{x}) = 0$.*

Based on Assumption 1, the non-ground-truth labels has lower relevance to the instance are prone to be chosen as IDCL due to the higher selection probabilities. To further study IDCLL, we present our generation process of IDCLs named "Min$k$" through the selection probability for each instance.

**Definition 1** (Min$k$). *We utilize a pre-trained model to obtain prediction probability $p(Y|X=\boldsymbol{x})$ for each instance $\boldsymbol{x}$, and calculate the selection probability $\bar{p}(\bar{Y}|Y=y, X=\boldsymbol{x})$ based on Assumption 1. Subsequently, we randomly select one complementary label for each instance from the labels with the $k$ highest selection probabilities.*

The rationale behind the above "Min$k$" stems from the belief that, although individuals may adhere to our assumption when selecting IDCLs, they actually do not consider the labels with high correlation to the ground-truth one. More importantly, this design allows us to generate various settings by varying the value of $k$ and recovery the uniform setting with $k = K-1$, thus simulating the diverse scenarios encountered in practice. Further analysis on the rationality can be found in Appendix E.

## 4.2 EMPIRICAL ANALYSIS OF EXISTING CLL METHODS

In this section, we first evaluate the performance of existing CLL methods under "Min3" setting. We select four baseline methods originally employed for solving uniform CLL: W_loss (Gao & Zhang, 2021), Forward (Yu et al., 2018), SCL_NL (Chou et al., 2020), and SCL_EXP (Chou et al., 2020). Table 1 presents the classification accuracy of these baseline methods on different benchmark datasets under the uniform and "Min3" settings, respectively. The results demonstrate a consistent decrease in accuracy for all the methods when switching from the uniform to "Min3" setting, confirming that most existing CLL methods are not well-suited for our instance-dependent setting.

We conducted experiments on the Kuzushiji-MNIST (KMNIST) dataset to further investigate the cause of the accuracy drop. When considering the existing CLL methods, most of them mathematically try to minimize the predictions of complementary label $p_{\boldsymbol{\theta}}(Y=\bar{y}|X=\boldsymbol{x})$ in one way or another. We calculate the average prediction of complementary label $p_{\boldsymbol{\theta}}(Y=\bar{y}|X=\boldsymbol{x})$ for SCL_NL over all instances in every epoch and Figure 1a presents SCL_NL effectively minimizes $p_{\boldsymbol{\theta}}(Y=\bar{y}|X=\boldsymbol{x})$ under "Min3" setting, which even lower than the value under the uniform setting. This phenomenon reveals the ineffectiveness of existing complementary losses: Although these complementary losses optimize their objectives, i.e., minimizing $p_{\boldsymbol{\theta}}(Y=\bar{y}|X=\boldsymbol{x})$, they still fail to make the ground-truth label prominent from the non-complementary labels.

To further explain this phenomenon, we first present the definition of *complementary label distribution* and introduce *share complementary label* hypothesis (Lin et al., 2023). Subsequently, we take an insight in why existing CLL methods demonstrate effectiveness under uniform setting.

**Definition 2** (Complementary label distribution). *The complementary label distribution refers to the distribution of complementary labels given the same ground-truth label, denoted as $p(\bar{Y}|Y=y)$. In addition, we use the entropy $H(\bar{Y}|Y)$ to characterize the dispersion of this distribution.*

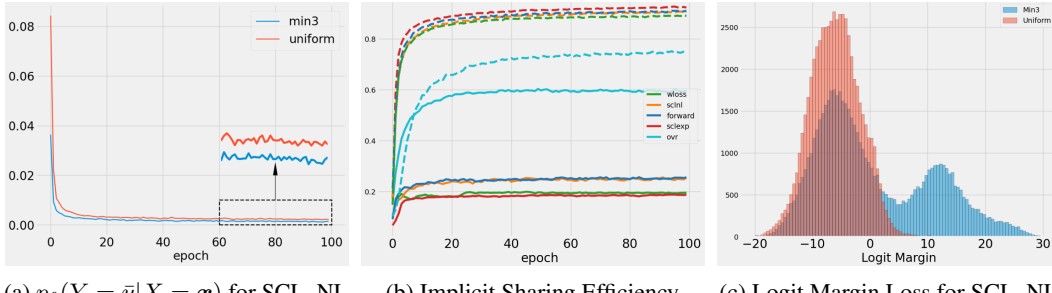

(a) $p_\theta(Y = \bar{y}|X = \boldsymbol{x})$ for SCL_NL    (b) Implicit Sharing Efficiency    (c) Logit Margin Loss for SCL_NL

Figure 1: (a) The prediction of complementary label $p_\theta(Y = \bar{y}|X = \boldsymbol{x})$; (b) Implicit sharing efficiency for various methods. Dotted line represents the uniform setting and the solid line represents the "Min3" setting; (c) Logit margin loss for SCL_NL under both uniform and "Min3" settings.

**Remark 1.** *Under the uniform setting, the complementary label distribution is assumed to be a uniform distribution over $K-1$ non-ground-truth labels. However, under the "Mink" setting, the complementary label distribution becomes relatively sparser, i.e., $H(\bar{Y}|Y)$ becomes smaller. As $k$ increases, the complementary label distribution approaches a uniform distribution, thus the "Min9" setting is equivalent to the uniform setting. More details can be found in Appendix F.*

**Hypothesis 1** (Share complementary label). *Instances that exhibit proximity in the feature space tend to share their complementary labels with each other. Under the uniform assumption, the complementary label distribution for each class is close to a uniform one, and each instance has access to all non-ground-truth labels as complementary labels shared by the neighboring instances.*

Next, we elucidate the significance of the ability to share complementary labels for CLL. We consider the logit margin loss (LML) $\ell_{\mathrm{LM}}$ in supervised classification as (Kanai et al., 2023):

$$\ell_{\mathrm{LM}} = \max_{j \neq y} g_j(\boldsymbol{x}) - g_y(\boldsymbol{x}). \tag{6}$$

When $\ell_{\mathrm{LM}} < 0$, the model correctly classifies the instance, whereas when $\ell_{\mathrm{LM}} > 0$, the model misclassifies the instance. Under the uniform setting, due to the share complementary label mechanism, existing CLL methods can potentially decrease the logit of all non-ground-truth label, i.e., $g_j(\boldsymbol{x}), \forall j \neq y$, thereby making $\ell_{\mathrm{LM}}$ small enough to correctly identify ground-truth label. However, the sparsity of the complementary label distribution under the instance-dependent setting diminishes the ability of existing CLL methods to share complementary labels, which makes the classifier hardly distinguishes the ground-truth label from the non-complementary labels. In order to validate our conjecture, we borrow the implicit sharing efficiency (ISE) from Lin et al. (2023):

$$\mathrm{ISE} = 1 - \frac{1}{n} \sum_{i=1}^{n} \frac{K-1}{K-2} \sum_{j \notin \{\bar{y}_i, y_i\}} p_{\boldsymbol{\theta}}(Y = j|X = \boldsymbol{x}_i). \tag{7}$$

This metric quantifies the decrease in the model's confidence regarding the unseen CLs. If implicit sharing helps identify all complementary labels, then $p_{\boldsymbol{\theta}}(Y = j|X = \boldsymbol{x}_i)$ will become zero, resulting in an ISE of one. In the absence of implicit sharing among instances, $p_{\boldsymbol{\theta}}(Y = j|X = \boldsymbol{x}_i)$ will be average $\frac{1}{K-1}$ and the ISE becomes zero. We calculate ISE during the training process for various CLL methods under both the "Min3" setting and the uniform setting. As shown in Figure 1b, different colors denote different CLL methods, with the dotted line representing ISE under the uniform setting and the solid line representing the "Min3" setting, respectively. Figure 1b reveals a substantial decrease in ISE for the existing CLL methods under the "Min3" setting compared to the uniform one. Furthermore, we present histograms of $\ell_{\mathrm{LM}}$ for SCL_NL under both uniform and "Min3" settings in Figure 1c, which clearly illustrates that SCL_NL exhibits smaller $\ell_{\mathrm{LM}}$ values under the uniform setting compared to the "Min3" setting. This observation solidifies the notion that greater ability of sharing complementary labels leads to smaller $\ell_{\mathrm{LM}}$, and consequently, improves the identification of potential ground-truth labels. From the empirical results, it is evident that existing CLL methods experience a diminished capacity to share complementary labels under the instance-dependent settings, which results in smaller $\ell_{\mathrm{LM}}$ and hinders their ability to effectively identify the ground-truth label.

## 5 METHODOLOGY

### 5.1 COMPLEMENTARY LOGIT MARGIN LOSS

Our empirical analysis highlights the connection between large $\ell_{\mathrm{LM}}$ and label disambiguation. However, in CLL problem, we are given only *single* complementary label for each instance and $\ell_{\mathrm{LM}}$ is underdetermined. In order to enhance ISE and identify the ground-truth label, we instead consider a novel complementary logit margin loss (CLML) $\bar{\ell}_{\mathrm{LM}}$ as:

$$\bar{\ell}_{\mathrm{LM}} = g_{\bar{y}}(\boldsymbol{x}) - \max_{j \neq \bar{y}} g_j(\boldsymbol{x}), \tag{8}$$

and define complementary logit margin (CLM) as $\max_{j \neq \bar{y}} g_j(\boldsymbol{x}) - g_{\bar{y}}(\boldsymbol{x})$. If $\bar{\ell}_{\mathrm{LM}} > 0$, the model misclassifies instance, while $\bar{\ell}_{\mathrm{LM}} < 0$ does not mean correct classification. However, $\bar{\ell}_{\mathrm{LM}}$ of correct classification should be negative and small, which suggests to penalize small CLM.

By decreasing CLML, we should minimizes $g_{\bar{y}}(\boldsymbol{x})$, which aligns with the existing CLL methods. However, CLML also takes into account $\max_{j \neq \bar{y}} g_j(\boldsymbol{x})$ and aims to maximize it. According to Assumption 1, IDCL tends to be the least relevant label to the instance, while the ground-truth label is typically the most relevant one and has the largest logit. This allows instances that exhibit proximity in the feature space to obtain not only the complementary label from neighbors, but also their ground-truth labels, which enhance the ISE and the ability to identify the ground-truth label.

### 5.2 COVR LOSS FOR IDCLL

For the purpose to increase CLML, it is essential to design a loss function that penalizes the small complementary logit margins. The CLML is a straightforward choice as the loss function. However, it exclusively considers the pair of the largest logit $g_j(\boldsymbol{x})(j \neq \bar{y})$ and the logit for the complementary label $g_{\bar{y}}(\boldsymbol{x})$, leading to a significant loss of information. In order to take all the logits into consideration, we utilize the Complementary One-Versus-Rest Loss (COVR) (Ishida et al., 2017):

$$\bar{\ell}_{\mathrm{COVR}}(y, \boldsymbol{g}(\boldsymbol{x})) = \ell(-g_{\bar{y}}(\boldsymbol{x})) + \frac{1}{K-1} \sum_{j \neq \bar{y}} \ell(g_j(\boldsymbol{x})). \tag{9}$$

In particular, we set $\ell(z) = \log(1 + e^{-z})$ and the COVR takes the form of:

$$\bar{\ell}_{\mathrm{COVR}}(y, \boldsymbol{g}(\boldsymbol{x})) = \log(1 + e^{g_{\bar{y}}(\boldsymbol{x})}) + \frac{1}{K-1} \sum_{j \neq \bar{y}} \log(1 + e^{-g_j(\boldsymbol{x})}). \tag{10}$$

We should point out that this novel form of COVR has not been proposed in previous work. As illustrated in Figure 1b, the implicit sharing efficiency of COVR demonstrates a substantial enhancement in its capacity to share complementary labels, allowing to better identify the ground-truth label. However, it is worth noting that ISE of COVR diminishes under the uniform setting. This decline is attributed to the fact that complementary labels obtained in the uniform setting are not necessarily the least relevant labels. Consequently, using COVR to decrease CLML may lead to overfitting. We also derived an estimation error bound for COVR to justify its convergence: To what extent, the empirical complementary risk minimization leads to the expected complementary risk minimization. The details of the estimation error bound can be found in Appendix C.

### 5.3 THEORETICAL ANALYSIS FOR COVR

To illustrate the effectiveness of COVR in decreasing CLML, we conduct a theoretical comparison between COVR and SCL_NL. First, we observe that COVR is the upper bound of SCL_NL:

**Theorem 1.** *If we consider COVR and SCL_NL, we can have*

$$0 \leq \bar{\ell}_{\mathrm{SCL\_NL}}(\bar{y}, \boldsymbol{g}(\boldsymbol{x})) \leq \bar{\ell}_{\mathrm{COVR}}(\bar{y}, \boldsymbol{g}(\boldsymbol{x})), \forall(\boldsymbol{x}, \bar{y}) \in \{(\boldsymbol{x}_i, \bar{y})\}_{i=1}^n. \tag{11}$$

*When $g_{\bar{y}}(\boldsymbol{x}) \to -\infty$ and $g_j(\boldsymbol{x}) \to +\infty, \forall j \neq \bar{y}$, we have $\bar{\ell}_{\mathrm{COVR}}(\bar{y}, \boldsymbol{g}(\boldsymbol{x})) \to 0$, and then $\bar{\ell}_{\mathrm{SCL\_NL}}(\bar{y}, \boldsymbol{g}(\boldsymbol{x})) \to 0$.*

For IDCLL, $\bar{\ell}_{\mathrm{COVR}}(\bar{y}, \boldsymbol{g}(\boldsymbol{x}))$ is consistently larger or at least equal to the baseline $\bar{\ell}_{\mathrm{SCL\_NL}}(\bar{y}, \boldsymbol{g}(\boldsymbol{x}))$. Furthermore, as $|\bar{\ell}_{LM}|$ increases towards infinity, both $\bar{\ell}_{\mathrm{COVR}}(\bar{y}, \boldsymbol{g}(\boldsymbol{x}))$ and $\bar{\ell}_{\mathrm{SCL\_NL}}(\bar{y}, \boldsymbol{g}(\boldsymbol{x}))$ converge towards zero asymptotically. Consequently, we anticipate that COVR penalize the large CLML more strongly than SCL_NL.

In addition, we further explore the impact of COVR on CLML to behavior of CLM by the problem:

$$\min_{\boldsymbol{g}} \bar{\ell}(\bar{y}, \boldsymbol{g}), \tag{12}$$

where $\bar{\ell}$ is set to $\bar{\ell}_{\mathrm{COVR}}$ or $\bar{\ell}_{\mathrm{SCL\_NL}}$, and $\boldsymbol{g} \in \mathbb{R}^K$ is the logit vector for a instance $\boldsymbol{x}$. To dissect the training dynamics of Eq. (12), we use the following assumption (Kanai et al., 2023):

**Assumption 2.** *The logit vector $\boldsymbol{g}$ follows the following gradient flow to solve Eq.(12):*

$$\frac{d\boldsymbol{g}}{dt} = -\nabla_{\boldsymbol{g}} \bar{\ell}(\bar{y}, \boldsymbol{g}), \tag{13}$$

*where $t$ is the time step of training. We assume that $\boldsymbol{g}$ is initialized to zeros $\boldsymbol{g} = \boldsymbol{0}$ at $t = 0$ .*

Eq. (13) serves as a continuous approximation of the gradient descent equation $\boldsymbol{g}^{\tau+1} = \boldsymbol{g}^{\tau} - \eta \nabla_{\boldsymbol{g}} \bar{\ell}$ and aligns with it as the learning rate $\eta$ approaches zero. It is a commonly used method to analyze the training dynamics (Kunin et al., 2021; Elkabetz & Cohen, 2021).

Under Assumption 2, we obtain the following two lemmas for the logits in the training of Eq.(12):

**Lemma 1.** *If we use $\bar{\ell}_{\mathrm{COVR}}(\bar{y}, \boldsymbol{g})$ in Eq.(12), the $j$-th logit $g_j$ at time $t$ is*

$$g_j(t) = \begin{cases} -t - 1 + W(e^{t+1}) & j \neq \bar{y}, \\ \frac{1}{K-1}t + 1 - W(e^{\frac{1}{K-1}t+1}) & j = \bar{y}, \end{cases} \tag{14}$$

*where $W$ is Lambert $W$ function, which is a function satisfying $x = W(xe^x)$ (Corless et al., 1996).*

**Lemma 2.** *If we use $\bar{\ell}_{\mathrm{SCL\_NL}}(\bar{y}, \boldsymbol{g})$ in Eq. (12), the $j$-th logit $g_j$ at time $t$ is*

$$g_j(t) = \begin{cases} \frac{1}{K-1}t + \frac{K-1}{K} - \frac{1}{K}W[(K-1)e^{\frac{K}{K-1}t+K-1}] & j \neq \bar{y}, \\ -t - \frac{(K-1)^2}{K} + \frac{K-1}{K}W[(K-1)e^{\frac{K}{K-1}t+K-1}] & j = \bar{y}. \end{cases} \tag{15}$$

These two lemmas provide insights into the trajectories of the logit vectors during the minimization of $\bar{\ell}_{\mathrm{COVR}}(\bar{y}, \boldsymbol{g}(\boldsymbol{x}))$ and $\bar{\ell}_{\mathrm{SCL\_NL}}(\bar{y}, \boldsymbol{g}(\boldsymbol{x}))$, respectively. Both methods decrease the complementary label logit $g_{\bar{y}}$ and increase the logits for non-complementary labels, but they exhibit different rates of update. From the above lemmas, we derive the trajectory of the complementary logit margins:

**Theorem 2.** *Complementary logit margin loss for the logit vector $\boldsymbol{g}^{\mathrm{COVR}}$ in the minimization of $\bar{\ell}_{\mathrm{COVR}}(\bar{y}, \boldsymbol{g}(\boldsymbol{x}))$ and logit vector $\boldsymbol{g}^{\mathrm{SCL\_NL}}$ in the minimization of $\bar{\ell}_{\mathrm{SCL\_NL}}(\bar{y}, \boldsymbol{g}(\boldsymbol{x}))$ at time t are*

$$\bar{\ell}_{\mathrm{LM}}(\boldsymbol{g}^{\mathrm{COVR}}(t)) = -\frac{K}{K-1}t - 2 + W(e^{t+1}) + W(e^{\frac{1}{K-1}t+1}), \tag{16}$$

$$\bar{\ell}_{\mathrm{LM}}(\boldsymbol{g}^{\mathrm{SCL\_NL}}(t)) = -\frac{K}{K-1}t - K + 1 + W[(K-1)e^{\frac{K}{K-1}t+K-1}]. \tag{17}$$

*For large t, they can be approximated by*

$$\bar{\ell}_{\mathrm{LM}}(\boldsymbol{g}^{\mathrm{COVR}}(t)) \approx -\log[(\frac{1}{K-1}t + 1)(t+1)], \tag{18}$$

$$\bar{\ell}_{\mathrm{LM}}(\boldsymbol{g}^{\mathrm{SCL\_NL}}(t)) \approx -\log[\frac{K}{(K-1)^2}t + 1 + \log(K-1)^{\frac{1}{K-1}}]. \tag{19}$$

*Then, we have $\lim_{t \to \infty} \frac{\bar{\ell}_{\mathrm{LM}}(\boldsymbol{g}^{\mathrm{COVR}}(t))}{\bar{\ell}_{\mathrm{LM}}(\boldsymbol{g}^{\mathrm{SCL\_NL}}(t))} = 2$ for any fixed $K$.*

Theorem 2 elucidates the disparity in the trajectories of CLML between COVR and SCL_NL under Assumption 2 and shows that SCL_NL does not decrease the CLML as small as COVR for sufficiently large $t$. As a result, COVR is proved to be more effective and efficient at decreasing the large CLML.

Table 2: Classification accuracy (mean±std) of various methods on benchmark datasets under "Min3" setting. The symbol • indicates the statistically significant superiority (at a significance level of 0.05).

| | MNIST | Kuzushiji-MNIST | Fashion-MNIST | CIFAR-10 | SVHN |
|---|---|---|---|---|---|
| PRODEN(PLL) | $63.84 \pm 1.12$ • | $46.37 \pm 0.74$ • | $40.74 \pm 3.77$ • | $40.72 \pm 1.69$ • | $33.05 \pm 1.06$ • |
| W_Loss | $46.19 \pm 4.56$ • | $46.73 \pm 3.94$ • | $42.61 \pm 6.14$ • | $45.57 \pm 0.63$ • | $38.46 \pm 3.13$ • |
| Forward | $51.82 \pm 6.81$ • | $53.69 \pm 4.50$ • | $39.54 \pm 0.13$ • | $45.12 \pm 2.03$ • | $31.43 \pm 2.86$ • |
| SCL_NL | $55.56 \pm 6.91$ • | $51.33 \pm 1.76$ • | $39.53 \pm 0.08$ • | $38.40 \pm 0.34$ • | $32.97 \pm 2.18$ • |
| SCL_EXP | $33.78 \pm 4.03$ • | $42.78 \pm 4.20$ • | $37.06 \pm 1.60$ • | $30.78 \pm 0.94$ • | $20.59 \pm 0.06$ • |
| NN | $60.55 \pm 4.88$ • | $37.67 \pm 2.24$ • | $39.15 \pm 2.22$ • | $34.03 \pm 1.51$ • | $33.67 \pm 2.84$ • |
| PC | $83.94 \pm 2.20$ • | $64.99 \pm 1.44$ • | $45.10 \pm 1.15$ | $37.51 \pm 1.21$ • | $72.53 \pm 1.67$ |
| COVR (ours) | $\mathbf{87.52 \pm 0.02}$ | $\mathbf{67.31 \pm 1.53}$ | $\mathbf{47.41 \pm 0.60}$ | $\mathbf{63.21 \pm 0.18}$ | $\mathbf{74.81 \pm 1.19}$ |

Table 3: Classification accuracy (mean±std) of different CLL methods on CIFAR10 under different instance-dependent settings. The symbol • indicates the statistically significant superiority (at a significance level of 0.05).

| Method | CIFAR10 | | | | CLCIFAR10 |
|---|---|---|---|---|---|
| | CIFAR10_Min1 | CIFAR10_Min3 | CIFAR10_Min5 | CIFAR10_Min7 | |
| Forward | $30.48 \pm 3.47$ • | $45.12 \pm 2.03$ • | $49.18 \pm 0.27$ • | $65.54 \pm 1.97$ | $36.83 \pm 1.17$ • |
| SCL_NL | $30.32 \pm 3.34$ • | $38.40 \pm 0.34$ • | $48.71 \pm 0.45$ • | $\mathbf{66.01 \pm 1.92}$ | $37.81 \pm 2.21$ |
| SCL_EXP | $25.97 \pm 0.90$ • | $30.78 \pm 0.94$ • | $42.74 \pm 3.31$ • | $55.05 \pm 1.32$ • | $36.96 \pm 0.18$ • |
| PC | $63.88 \pm 0.54$ | $37.51 \pm 1.21$ • | $49.76 \pm 1.02$ • | $44.20 \pm 1.62$ • | $35.88 \pm 0.98$ • |
| COVR (ours) | $\mathbf{66.22 \pm 1.34}$ | $\mathbf{63.21 \pm 0.18}$ | $\mathbf{62.23 \pm 0.59}$ | $64.34 \pm 1.08$ | $\mathbf{38.29 \pm 0.21}$ |

# 6 EXPERIMENTS

## 6.1 SETUPS

**Datasets and Baselines** In our experiments, we adopt five widely used benchmark datasets including MNIST , Fashion-MNIST (FMNIST) , Kuzushiji-MNIST (KMNIST) , CIFAR-10, and SVHN . For MNIST, FMNIST and KMNIST, we train a convolutional neural network with two convolutional layers and two fully-connected layers for 100 epochs with batch size of 256. Adam optimizer (Kingma & Ba, 2014) is used with the learning rate$=1\times10^{-3}$, weight decay$=1\times10^{-4}$. For CIFAR-10 and SVHN, we train ResNet-18 (He et al., 2016) for 200 epochs with batch size of 128. SGD optimizer is used with the learning rate$=1\times10^{-1}$, weight decay$=5\times10^{-4}$. We choose six existing CLL methods to date, including W_Loss (Gao & Zhang, 2021), Forward (Yu et al., 2018), SCL_NL (Chou et al., 2020), SCL_EXP (Chou et al., 2020), NN (Ishida et al., 2019), PC (Ishida et al., 2017), and a partial label learning method PRODEN (Lv et al., 2020). To generate IDCLs for each dataset, we use MLP and ResNet-18 trained with ground-truth labels as the pre-trained models and follow the generation processes of IDCLs defined in Section 4. More experimental details can be found in Appendix D.

## 6.2 EXPERIMENTAL RESULTS

**Classification accuracy on benchmark datasets** Table 2 reports the classification accuracy of each CLL method on benchmark datasets under "Min3" setting. The best results are highlighted in bold and the symbol • indicates that our method is statistically superior to the comparing methods on each dataset (pairwise t-test at a significance level of 0.05). From Table 2, COVR demonstrates the best performance and significantly outperforms other existing CLL methods under "Min3" setting. Furthermore, we find that COVR exhibits a lower standard deviation compared to other baseline methods, indicating its improved consistency and stability under the instance-dependent setting.

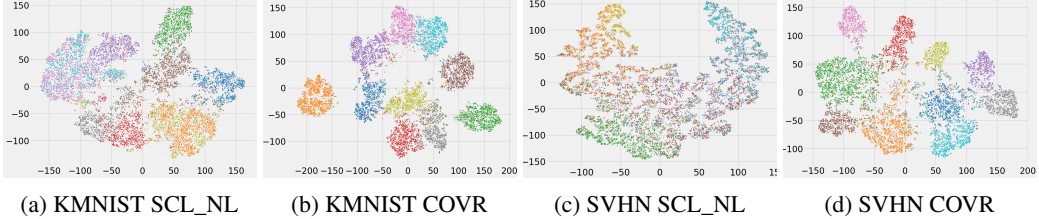

| (a) KMNIST SCL_NL | (b) KMNIST COVR | (c) SVHN SCL_NL | (d) SVHN COVR |

Figure 2: The t-SNE visualization of the image representation on KMNIST (left panel) and SVHN (right panel) under "Min3" setting. Different colors represent the corresponding classes.

**The effectiveness of COVR under different instance-dependent settings** To validate the effectiveness of COVR, we conducted experiments using the CIFAR10 with various instance-dependent settings, including "Min1", "Min5", and "Min7". A human-annotated CLL dataset CLCIFAR10 (Wang et al., 2023) (more details in Appendix H) is also considered. Table 3 presents the accuracy of different CLL methods across these instance-dependent settings on the CIFAR10 dataset. We observe that COVR consistently performs well under various instance-dependent settings, and as $k$ increases (becoming more uniform), COVR does not exhibit significant performance degradation. As Figure 1b illustrates, COVR exhibits a more stable performance across both uniform and instance-dependent settings. Moreover, our proposed approach demonstrates notable improvement compared to other methods when evaluated on the human-annotated CLCIFAR10 dataset. These results emphasize the effectiveness of our approach on the CLCIFAR10 dataset, further validating its superiority in practice.

**COVR learns more distinguishable representations** In Figure 2, we present a visualization of the image representations generated by the feature encoder using t-SNE (Torralba et al., 2008). Different colors represent distinct ground-truth labels. This analysis was conducted using the KMNIST and SVHN datasets under the "Min3" setting, respectively. We compare the t-SNE embeddings of two approaches: (a) The best-performing baseline method, i.e., SCL_NL, and (b) Our method, i.e., COVR. It is shown that on both datasets, representation learned by SCL_NL is indistinguishable, especially for the more complex dataset SVHN. Conversely, COVR consistently generates well-separated clusters and learns distinct representations on both datasets. This observation confirms that COVR allows the model to learn the feature that is more relevant to the instance (Lv et al., 2020; Wang et al., 2023).

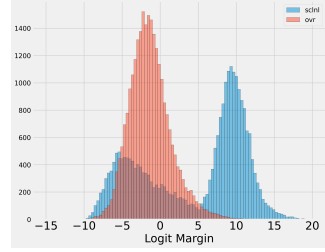

**COVR effectively increases the logit margin** In Figure 3, we display histograms of $\ell_{\mathrm{LM}}$ for both SCL_NL and COVR under the "Min3" setting on SVHN. Instances with $\ell_{\mathrm{LM}} > 0$ are misclassified. Figure 3 reveals that by reducing $\bar{\ell}_{\mathrm{LM}}$, COVR enhances the capacity to share complementary labels and subsequently decreases $\ell_{\mathrm{LM}}$. This improvement allows COVR to better identify potential ground-truth labels compared to SCL_NL, which validates our explanation.

Figure 3: $\ell_{\mathrm{LM}}$ for SCL_NL and COVR under "Min3".

## 7 CONCLUSION

In this paper, we for the first time introduce the instance-dependent assumption to complementary label learning and present the generation process for IDCL according to our assumption. Subsequently, we discover that directly applying existing CLL methods to IDCLL results in a poor performance and further empirically identify the underlying reason as the diminishing capacity to share complementary labels. To address this problem, we introduce CLML to enhance the share of complementary labels. We additionally propose COVR as the surrogate loss, and thoroughly investigate its advantages in theoretical and empirical under our instance-dependent setting.

Nevertheless, there are three limitations in our study. First, although we explore the share complementary label mechanism, COVR cannot explicitly utilize this mechanism, especially under uniform setting. Second, the generation process of IDCL presented in our work may not be the most realistic one. Third, the estimation of instance-dependent transition matrix may be a better solution for IDCLL. To overcome these limitations will shed light on the promising improvement of our current work.

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
