# OpenReview forum: "Rethinking One-vs-the-Rest Loss for Instance-dependent Complementary Label Learning"
_ICLR.cc/2024/Conference — ICLR 2024 Conference Withdrawn Submission_

### Official Review · Reviewer_uKaT · 2023-10-14

**Soundness:** 2 fair
**Presentation:** 3 good
**Contribution:** 2 fair
**Rating:** 5
**Confidence:** 4

**Summary:**

In this paper, the authors introduce instance-dependent CLL (IDCLL), where non-ground-truth labels that are less relevant to the instances are more likely to be selected as the complementary ones. To address this problem, the authors introduce CLML to enhance the share of complementary labels, and additionally propose COVR as the surrogate loss.

**Strengths:**

1.	The setting studied in this paper is interesting and useful.
2.	The experimental results justify the effectiveness of the proposed algorithm.

**Weaknesses:**

1.	The authors explain that the reason for bad performance of existing methods in IDCLL is “The sparser complementary label distribution under instance-dependent settings diminishes the capacity of existing CLL methods to effectively share complementary labels, resulting in small logit margin losses and making it challenging to disambiguate for the potential ground-truth labels.” I think this is somehow the observation. The authors try to justify this point by some empirical studies, which I think is not convincing. It would be better if the authors can reveal the intrinsic reason from the aspect of complementary label generation which is related to groundtruth label, namely starting from p(\bar{Y}=\bar{y}|Y=y, X=x). For example, the authors found that the sparser complementary label distribution will appear under IDCLL. But why? Maybe it is related to the label generation probability. In other words, I think the authors should make more insightful investigations on this setting.
2.	Although the proposed loss function is new for CLL, it is actually a very straightforward adaptation from (Ishida et al., 2017). Considering that there have been intensive researches on instance-dependent settings in other problems (e.g., label noise, semi-supervised learning, PU learning, etc.), and similar loss has been developed. I think the proposed loss in this paper is not that interesting nor sufficiently novel.
3.	The writing of this paper needs improvement. For example, in abstract, I cannot fully understand “capacity to share complementary labels”. Why sharing complementary labels? Who will share these labels? with whom?
4.	The experiments are only conducted on some synthetic datasets. It would be better if the authors can find some benchmark datasets that naturally follow the setting of IDCLL.

**Questions:**

I do not have specific questions on this paper.

---

> ### Author Response · Authors · 2023-11-14
> **We are very grateful for your time and effort in reviewing this submission.**
>
> We are very grateful for your time and effort in reviewing this submission. We have to admit that some possible improvements should be made to our paper. However, please note that we have indeed conducted the comparison with other CLL methods on a **human-annotated instance-dependent** CLL dataset CLCIFAR10 (real-world) in Table 3 on page 8.
>
> As we may need to make some revisions to this paper (including the explanations of the sparser complementary distribution under the instance-dependent setting than the uniform one, and more insights on rationale and effectiveness of the proposed CLML for IDCLL), we decide to withdraw this submission later.

---

### Official Review · Reviewer_dowv · 2023-10-28

**Soundness:** 3 good
**Presentation:** 3 good
**Contribution:** 2 fair
**Rating:** 5
**Confidence:** 3

**Summary:**

This paper introduces instance-dependent CLL (IDCLL), where non-ground-truth labels less relevant to the instances are selected as the complementary ones, which is said to be different from the previous CLL methods that the complementary label is sampled uniformly from all non-ground-truth labels, or from a biased probability depending on the ground-truth label. Further, it empirically demonstrates that existing methods perform bad under the instance-dependent setting, thus introduce complementary logit margin loss (CLML), and the complementary one-vs-the-rest loss (COVR) as the surrogate loss for CLML as well. Experiments on benchmark datasets verify its effectiveness.

**Strengths:**

This paper introduces a new instance-dependent CLL (IDCLL), which is different from the previous CLL settings, and attempt to demostate that the previous methods cannot work. Moveover, it proposes a new complementary logit margin loss (CLML) to solve it.

**Weaknesses:**

This paper attempts to demostate that the previous methods cannot work.  Why the complementary label distribution under the instance-dependent setting is sparse need more explain. This paper revises the logit margin loss (LML) to complementary logit margin loss (CLML), if it is indeed benifit to the correct classification? Moreover,  there is no comparison with more recent methods in the experiments.

**Questions:**

1.	It is said that an annotator tends to choose a label that is largely irrelevant to the instance to avoid mistaking the ground-truth label as the complementary one. Is the “irrelevant” non-ground-truth label consistent with the classifier prediction?
2.	If the complementary label is the most irrelevant one to instance, then nearby instances are more likely to share the same complementary label then uniform ones, why the complementary label distribution under the instance-dependent setting is sparse? There should be more explain.
3.	This paper revises the logit margin loss (LML) to complementary logit margin loss (CLML), which enlarges the margin between complementary label and the other labels, in this case, the model will assign the instance to complementary label with smaller probability, or misclassifies instance to the complementary label with smaller probability. However, does it have influence to the marge between ground-truth label and others? Or why it benefits the correct classification?
4.	In experiments, the proposed method is compared with the SOTA methods in instance-dependent CLL (IDCLL) setting, what about the other settings?
5.	Moreover, it is suggested to compared with more recent methods in the current two years.

---

> ### Author Response · Authors · 2023-11-14
> **We appreciate for your time and effort in reviewing this submission.**
>
> We appreciate for your time and effort in reviewing this submission. We have to admit that some possible improvements should be made to our paper. However, we have to point out that we have already conducted the comparisons in other CLL settings, i.e., **Uniform CL** in Table 1 on page 4 and Table 6 on page 22 (see the *Min9* case), and **Biased CL** in Table 7 on page 23.
>
> As we may need to make some revisions to this paper (including the explanations of the sparser complementary distribution under the instance-dependent setting than the uniform one, and more insights on rationale and effectiveness of the proposed CLML for IDCLL), we decide to withdraw this submission later.

---

### Official Review · Reviewer_8ujZ · 2023-10-31

**Soundness:** 2 fair
**Presentation:** 3 good
**Contribution:** 2 fair
**Rating:** 3
**Confidence:** 4

**Summary:**

This manuscript designs a loss for complementary label learning under the setting of instance-dependent complementary labels.

**Strengths:**

The CLL is interesting and the writing is easy to follow.

**Weaknesses:**

The methods compared in this paper are too old, even a little bit out of date. I am afraid that most of the state-of-the-art
methods are not included.

In Eq.(10), I would like to see the effect of replacing the loss \ell(z) with MSC or CCE.

Since cll is a special case of pll, methods' comparison should contains the state-of-the-art pll methods.

**Questions:**

``Existing methods exhibit a decline in their capacity to share complementary labels under the instance-dependent setting, resulting in small logit margins, thus difficult to identify ground-truth labels.'' Is there some evidence to verify this opinion? It will be more convincing if the authors can provide more analysis.

``Existing CLL methods assume that the complementary label is sampled uniformly from all non-groundtruth labels, or from a biased probability depending on the ground-truth label. However, these assumptions are normally unrealistic, for example, an annotator tends to choose a label that is largely irrelevant to the instance to avoid mistaking the ground-truth label as the complementary one.'' In my view, an annotator always selecting a label largely irrelevant to the instance is unrealistic, in other words, an annotator why not select the ground truth label instead of such a complementary label since it takes the same time-consuming.

---

> ### Author Response · Authors · 2023-11-14
> **We are grateful for your time and effort in reviewing this submission.**
>
> We are grateful for your time and effort in reviewing this submission. We have to admit that there are still some possible improvements should be made to our paper. However, we have conducted the comparison with one commonly used PLL algorithm (i.e., **PRODEN**, please see the first row of Table 2 on page 8 of our submission).
>
> As we may need to make some revisions to this paper (including the explanations of the sparser complementary distribution under the instance-dependent setting than the uniform one, and more insights on rationale and effectiveness of the proposed CLML for IDCLL), we decide to withdraw this submission later.